

# A Viterbi decoder and its hardware Trojan models: an FPGA-based implementation study

Varsha Kakkara[1],[*], Karthi Balasubramanian[1], B. Yamuna[1], Deepak Mishra[2], Karthikeyan Lingasubramanian[3] and Senthil Murugan[4],[*]

[1] Department of Electronics and Communication Engineering, Amrita School of Engineering, Amrita Vishwa Vidyapeetham, Coimbatore, Tamil Nadu, India
[2] Digital Communication Division (DCD), Optical and Digital Communication Group (ODCG), Satcom Navigation Payload Area (SNPA), Space Application Center (SAC), ISRO, Ahmedabad, Gujarat, India
[3] Electrical and Computer Engineering, University of Alabama, Birmingham, AL, USA
[4] Department of Electronics and Communication Engineering, Amrita School of Engineering, Amrita Vishwa Vidyapeetham, Amritapuri, Kerala, India
[*] These authors contributed equally to this work.

Corresponding author
Karthi Balasubramanian,
b_karthi@cb.amrita.edu

## ABSTRACT

Integrated circuits may be vulnerable to hardware Trojan attacks during its design or fabrication phases. This article is a case study of the design of a Viterbi decoder and the effect of hardware Trojans on a coded communication system employing the Viterbi decoder. Design of a Viterbi decoder and possible hardware Trojan models for the same are proposed. An FPGA-based implementation of the decoder and the associated Trojan circuits have been discussed. The noise-added encoded input data stream is stored in the block RAM of the FPGA and the decoded data stream is monitored on the PC through an universal asynchronous receiver transmitter interface. The implementation results show that there is barely any change in the LUTs used (0.5%) and power dissipation (3%) due to the insertion of the proposed Trojan circuits, thus establishing the surreptitious nature of the Trojan. In spite of the fact that the Trojans cause negligible changes in the circuit parameters, there are significant changes in the bit error rate (BER) due to the presence of Trojans. In the absence of Trojans, BER drops down to zero for signal to noise rations (SNRs) higher than 6 dB, but with the presence of Trojans, BER doesn't reduce to zero even at a very high SNRs. This is true even with the Trojan being activated only once during the entire duration of the transmission.

## INTRODUCTION

The entry of connected technologies into the realms of Internet of Things (IoT) and cyber physical systems (CPS) has made it imperative for communications systems to be protected from possible threats. These threats can arise from both software externals and hardware internals. While considerable emphasis is being given to software level threats, in this work we focus on the hardware level threats. The hardware of a communication system can be compromised if its design is exposed so that it can be modified or duplicated.

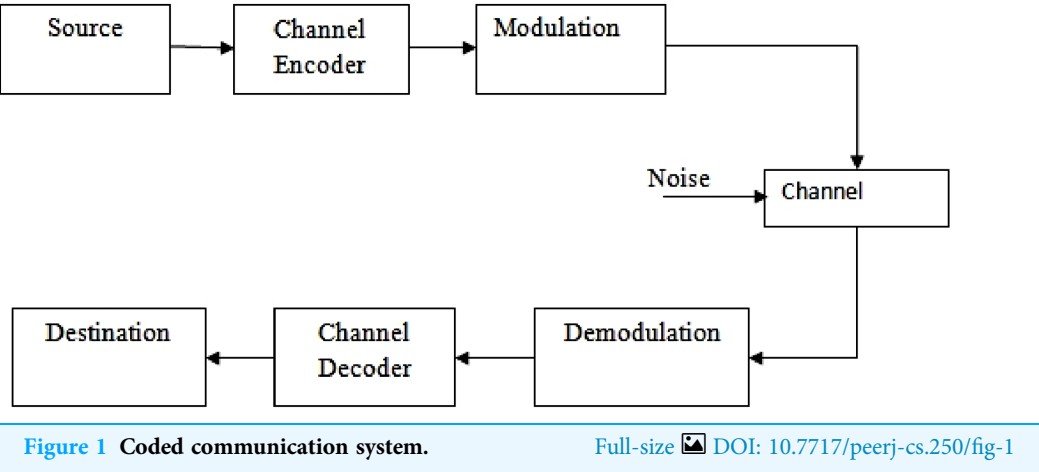

**Figure 1 Coded communication system.**

This allows an adversary to deteriorate the performance of a communication system and expose the system to attacks. This makes understanding of the hardware level threats significant. In this work, we focus on the effect of one such threat called hardware Trojans, on coded communication systems that use a Viterbi decoder as the error correcting unit.

## Overview of coded communication system

Design of efficient coder–decoder for error control has received increased interest in recent years. This is due to the fact that all digital transmission and storage requires error control strategy to ensure reliability. Information symbols from a source are encoded by the addition of controlled redundancy. Convolutional codes and block codes are the broad classification of error control codes. An error control decoder makes the best estimate of the transmitted codeword by making use of the redundancy added at the encoder. The transmitted codewords are encoded information symbols that are subject to errors, in the process of transmission through noisy communication channels. These transmitted codewords can be decoded with as low bit error rate (BER) as possible for transmission rates upto the channel capacity (*Sweeney, 2002*). The block level representation of a coded communication system is shown in Fig. 1.

## Hardware Trojans

In the current scenario of integrated circuits (ICs) manufacturing, a globalized business model has emerged where ICs are manufactured in foundries that are distributed in various parts of the world. A hardware Trojan is a malicious stealthy modification that leads to malfunctioning of the system (*Colins, 2007*). Such modifications in the system provides a back door entry for the Trojans. The three main categories of hardware Trojans are based on their action, physical and activation characteristics (*Chakraborty, Narasimhan & Bhunia, 2009*; *Tehranipoor & Koushanfar, 2010*; *Karri et al., 2010*, *Banga et al., 2008*; *Ranjani & Devi, 2017*). The physical characteristics category describes the various hardware manifestations of Trojans according to their shape and

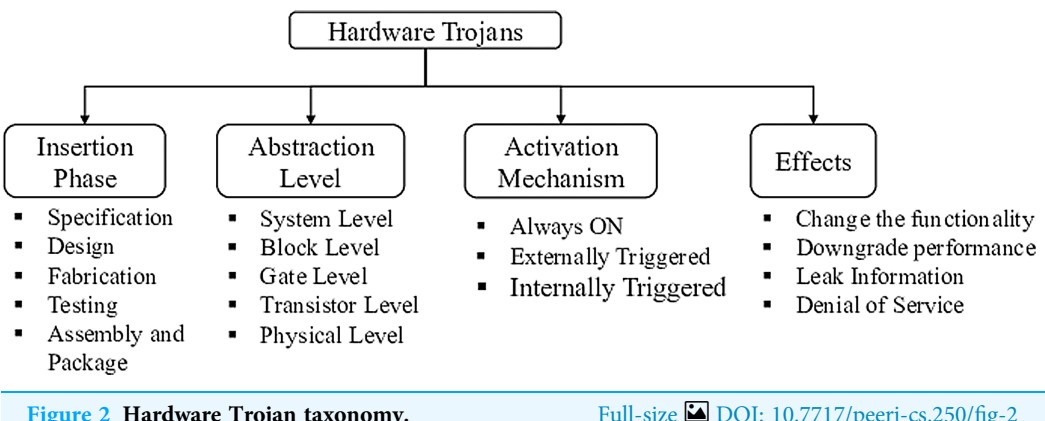

**Figure 2** **Hardware Trojan taxonomy.**

size; the activation characteristics describe the conditions that activate the Trojans and action characteristics refer to the behavior of the Trojans. Figure 2 gives the classification of Trojans based on insertion phase, abstraction level, activation mechanism and effects.

Hardware Trojans can be inserted at different stages of IC design cycle, while the most prevalent phases are design and fabrication. Likewise, Trojans can be realized at different levels of IC design abstraction and can be designed to get triggered internally by specific states of the system, or externally through any communication medium. The former can be stealthy based on the occurrence of the problem states, while the latter will be untraceable in test phase because it is not triggered internally. Regarding the effect of Trojans on the affected system, they are generally designed by the adversary to change functionality or leak sensitive information or deny service during critical instances or compromise the communication system and reduce the reliability of the design (*Karri et al., 2010*).

There are numerous post manufacturing techniques for detecting Trojans but a single technique is difficult to be devised for detecting Trojans universally. Side channel and logic testing form the two classical Trojan detection techniques (*Narasimhan et al., 2013*). In these two methods, a golden circuit is used to compare with outputs of the circuit under test. Typically, Trojans are devised to activate rarely to escape logic testing and evade detection. They also possess small physical characteristics to evade side channel based testing.

### Trojan modeling

Trojans are generally modeled for the specific design of interest that they intended to disrupt. Examples of Trojan benchmark circuits aimed at infecting systems like advanced encryption system, serial interface RS232, Ethernet MAC, 8051 and PIC microcontrollers can be found at (*TrustHUB, 2019*). These circuit models have been widely used to study the effectiveness of Trojan infection and to design measures to thwart them. To study the Trojan effect on other systems, custom Trojan models are designed. A few

studies that have been done in recent years on a system level using custom Trojan models are listed below:

1. *Saeidi & Garakani (2016)*: Multiple hardware Trojans have been designed for a 256 × 128 array of six transistor SRAM block, that either corrupt the output or modify the delay and duty cycle of the enable signals. The Trojans are trigged by an address sequence that is not generally produced in conventional testing methodology, thereby helping them to evade detection by conventional SRAM testing.

2. *Tiwari et al. (2019)*: A hardware Trojan model has been proposed for launching denial of service attack to on-chip multicast routing algorithms. The Trojan is modeled to use the on-chip temperature sensor information to identify suitable nodes and launch attack on multicast data packets.

3. *Liu et al. (2016)*: The design and custom silicon implementation of secret key leaking Trojans present in the ultrawide band transmitter of a wireless cryptographic IC has been presented. The Trojan circuit leaks the encryption key without disrupting the normal operation. This is achieved by hiding the key in the power amplitude and frequency margins that are acceptable due to process variations.

4. *Kumar et al. (2018)*: Novel hardware Trojans are proposed that induces denial of service and performance degradation in a Network on Chip. The Trojan is triggered by a complex bit pattern generated from input messages, intended toward misleading the packets away from the destination address.

5. *Subramani et al. (2019)*: Hardware Trojan attack is modeled by modifying the encoder block of a 802.11 a/g transmitter. This is accomplished by hijacking some of the legitimately encoded bits and substituting with rogue bits.

### Trojan modeling of channel decoders

Channel decoders are a quintessential part of any coded communication system. They are soft targets for Trojan attacks and can be embedded with malicious blocks for the following reasons: (*Hemati, 2016*).

1. They have a direct interface with the outside world that make them susceptible to being hijacked.

2. They process noisy information that makes it impossible for a even a perfectly functional decoder to be successful all the time. Hence a Trojan affected system may easily claim false failures and masquerade its real purpose.

3. Brute force approach of running all test cases to identify malicious activity is not practical with even a medium size block length since the number of input and output combinations will be huge.

In spite of the fact that channel decoders are highly susceptible to Trojan attacks, the effects of Trojans on them hasn't been explored in literature. *Hemati (2016)* have proposed the use of stochastic techniques at a system level for mitigating Trojan effects in a channel decoder but an RTL level analysis is missing. Our work involves the proposal and

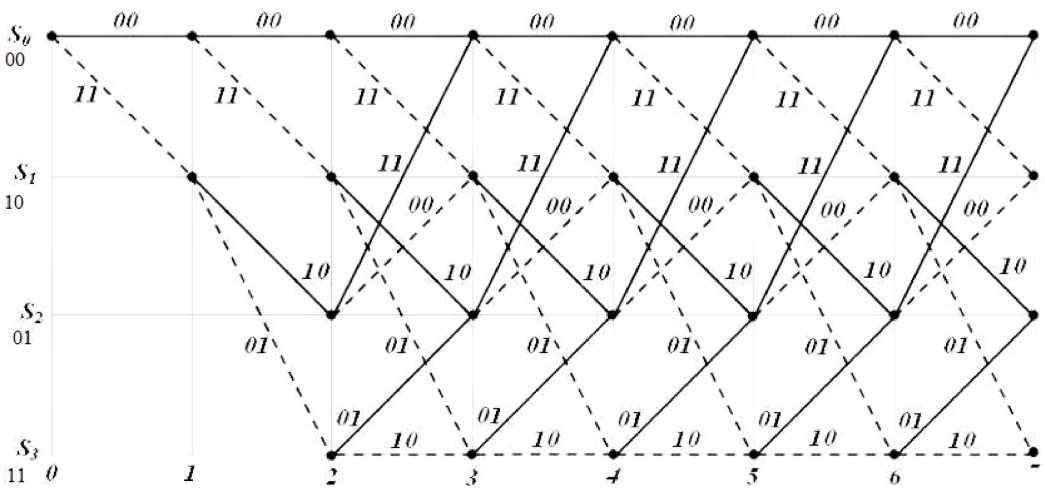

**Figure 3 Trellis structure of the convolutional encoder.**

analysis of possible Trojans on a specific channel decoder namely, the Viterbi decoder. The work is concentrated toward RTL design of a Viterbi decoder and possible Trojans that may potentially compromise the communication system and reduce the reliability of the decoder.

Viterbi algorithm is widely used for decoding convolutional codes since it achieves maximum likelihood estimate of the convolutionally coded transmitted sequence (*Forney, 1973*). A low BER can be achieved by a Viterbi decoder (*Viterbi, 1967*). However the presence of Trojans can affect the performance of the decoder significantly. This has been demonstrated with a proof of concept in our earlier work (*Aravind et al., 2018*). Trojan models were proposed and behavioral modeling studies at the algorithmic level showed that the BER performance of convolution decoder using the Viterbi algorithm is degraded due to the presence of the hardware Trojans. The current work extends this proof of concept to a RTL level circuit design of the decoder and the Trojan activities. A practical implementation of the Viterbi decoder is achieved and the Trojan effects on the system is analyzed.

The article is organized as follows. The Viterbi decoder section details the reader about the Viterbi algorithm with a suitable example. This is followed by the section on the hardware design of the decoder for FPGA implementation. Results from simulation and FPGA implementation of the decoder are discussed after that, followed by the section on the design of the Trojans. Results and discussions on the Trojan based design are presented and the article then concludes with references to possible future work.

## VITERBI DECODER

The $n$ encoded output in a $(n, k, m)$ convolutional code depends on the $k$ present input blocks as well the $m$ past input blocks. A memory $m$ sequential circuit is used for realizing the convolutional encoder. The trellis diagram of the rate half, $m = 2$, convolutional encoder is shown in Fig. 3. The corresponding state diagram of the trellis is shown in Fig. 4.

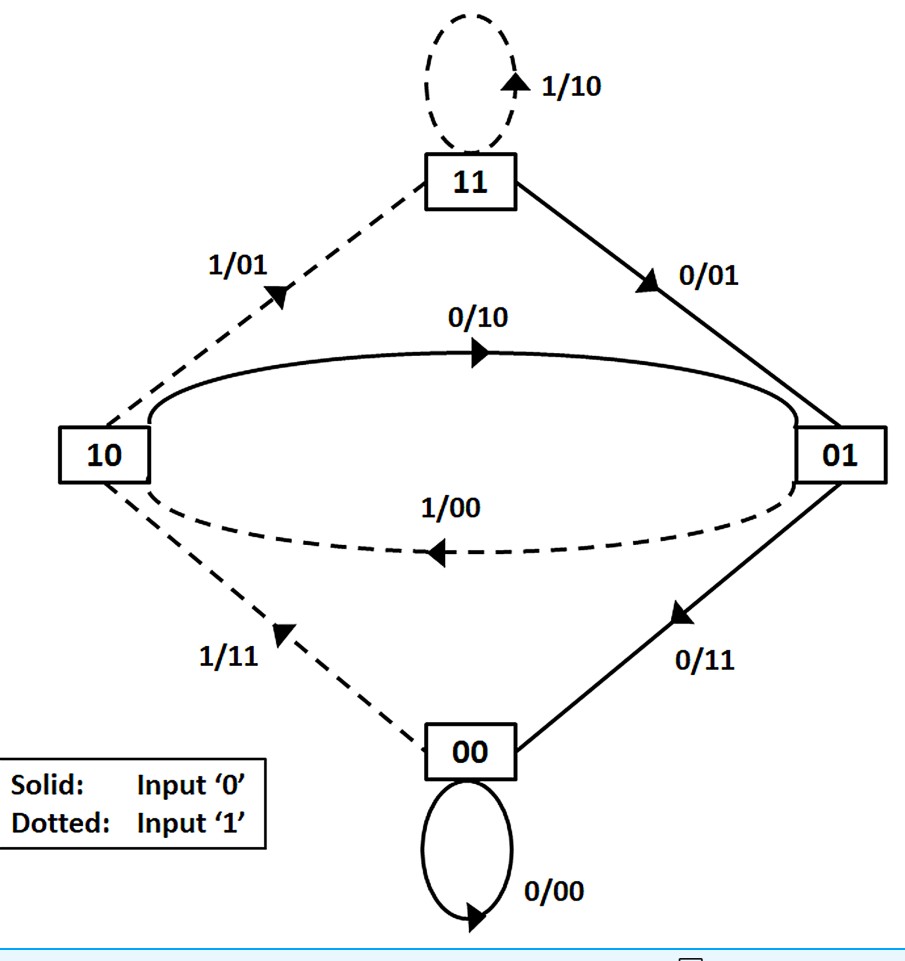

**Figure 4  State transitions of the encoder.**

The state transitions and the outputs reached in response to changes in input are represented in the state diagram. In the trellis diagram the same information is described in stages which represent the different time instants. With states representing the past memory contents of the encoder and branches representing the state transitions, the trellis path traced by the Viterbi decoder is a sequence of branches. The input message bit corresponding to each branch in the sequence of branches represent the decoded message sequence. By adding terminating zeros to the message sequence it is ensured that the decoder always starts in an all-zero initial state for decoding a message sequence. A detailed example showing the decoding procedure may be found in the Supplemental Document.

## FPGA BASED VITERBI DECODER DESIGN

The Viterbi decoder was designed in Verilog and implemented on a Xilinx Zybo-Z7010 board. Figure 5 shows the top level implementation structure that includes the core decoder block, a single port block RAM (BRAM) of size $8 \times 1{,}024$ to store the input data

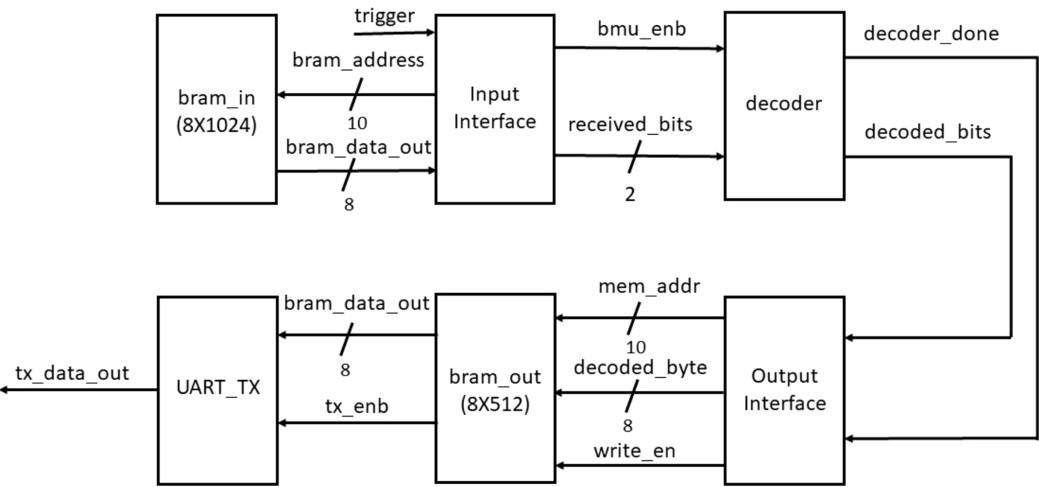

**Figure 5 Block level diagram for FPGA implementation of Viterbi decoder.**

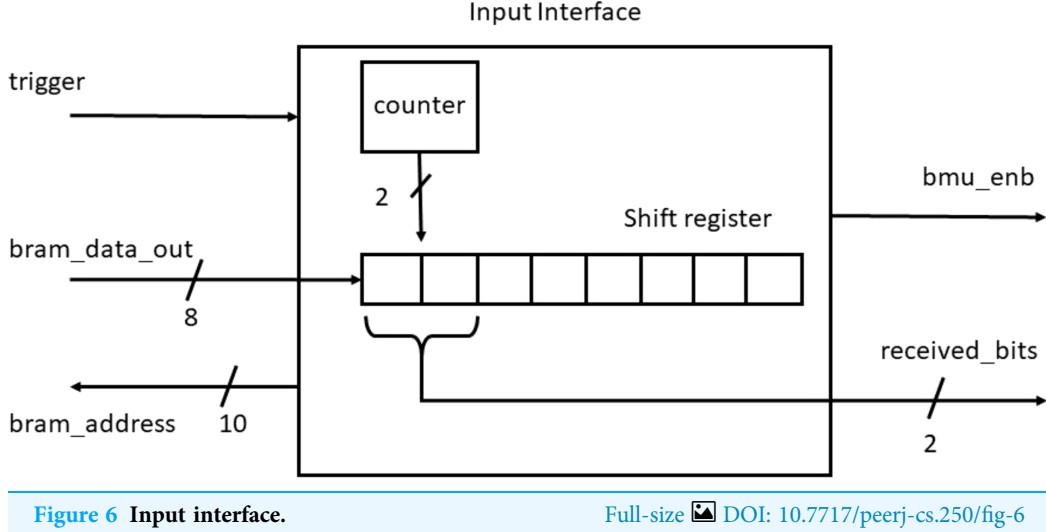

**Figure 6 Input interface.**   

and a single port block RAM of size 8 × 512 to store the decoded data. Along with these, an universal asynchronous receiver transmitter (UART) transmitter module is also integrated to monitor the decoded data on a PC.

## Input interface

The noise added encoded message is stored in the input BRAM and data is transferred to the decoder block through the input interface. The interface logic unit consists of a counter and an eight bit shit register as shown in Fig. 6.

It reads the data byte-wise from the BRAM and transmits two bits per clock cycle to the decoder block for further processing. At every clock cycle two LSB bits are shifted out to the branch metric unit (BMU) block. After every four clock cycles, the subsequent BRAM location is read and processed similarly.

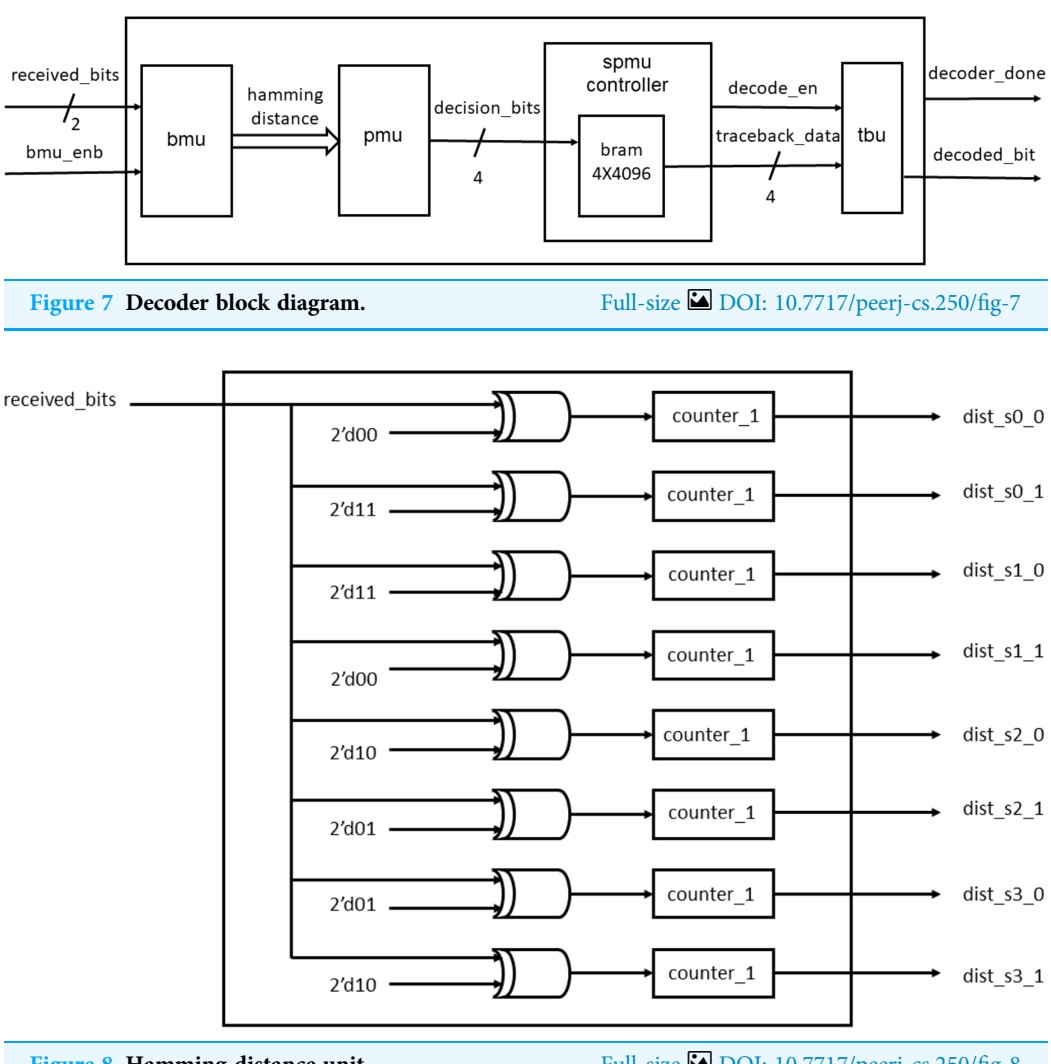

**Figure 7  Decoder block diagram.**

**Figure 8  Hamming distance unit.**

## Decoder design

Figure 7 shows the top level block diagram of the decoder consisting of the BMU, path metric unit (PMU), survivor path memory unit (SPMU), spmu_controller and the trace back unit (TBU).

### *Branch metric and path metric units*

The BMU calculates the Hamming distance between the received frame and the branch word while the PMU performs add-compare-select (ACS) calculations as described in *Middya & Dhar (2016)*. Figures 8 and 9 show the blocks used for the Hamming distance calculation and the ACS units.

During every clock cycle, BMU calculates the eight branch metrics corresponding to the two transitions of each of the four states. The branch metrics are then passed on to the PMU that updates each state with the least path metric corresponding to each of the states and also stores the corresponding path leading to it in the form of a "decision bit," one for each of the four states at every time instant. It can be seen from the trellis diagram in

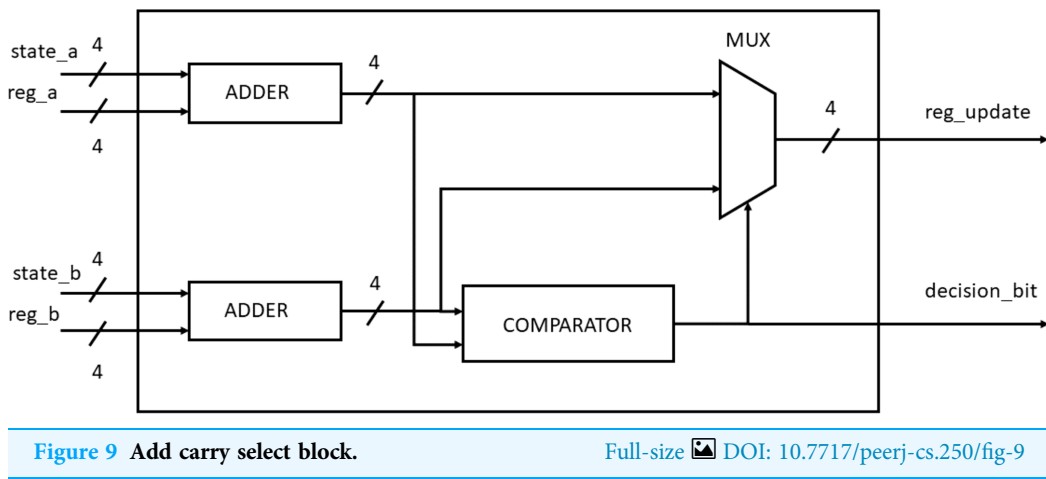

**Figure 9  Add carry select block.**     

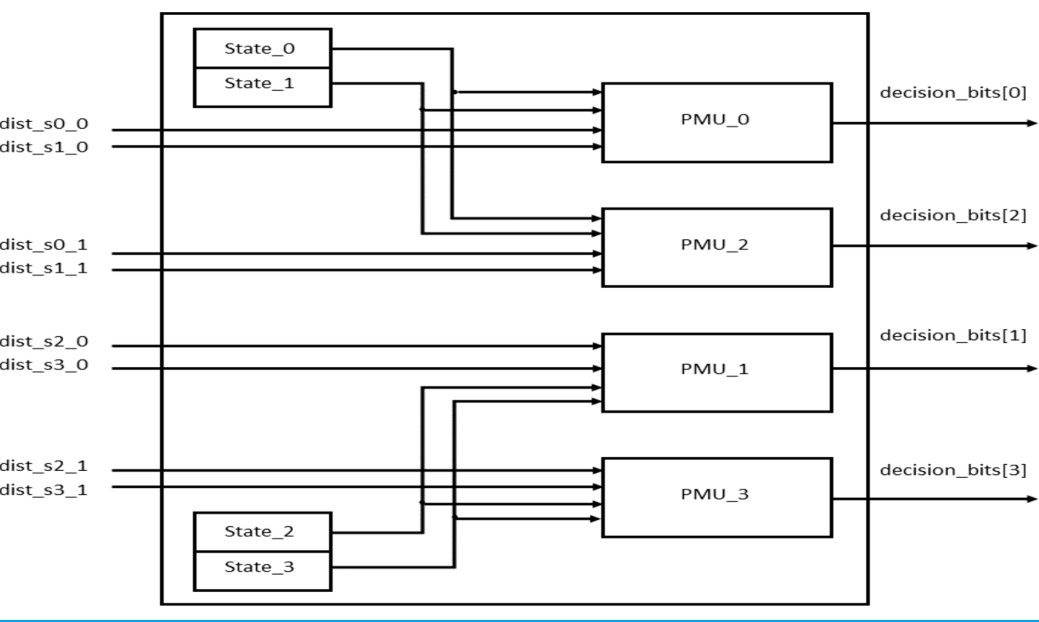

**Figure 10  PMU blocks for all the four states.**     

Fig. 3 that each state has two incoming paths. The decision bit is set to "0" if the state is reached from the top branch while it is set to "1" if it is reached from the bottom branch. Figure 10 shows the block diagram of the PMU block to update the path metrics and for decision bit calculations for the four states.

### Survivor path memory unit and trace back unit

Survivor path memory is designed as a single port BRAM that stores the decision bit values of all states at each of the time instants. The structure of the SPMU is the same as the trellis structure shown in Fig. 3. For the input data size of 1,024 bytes, the number of decision bits generated would be 4,096 for every state. Thus a $4 \times 4,096$ BRAM is used as the SPMU. The SPMU_update module reads the decision bits from the PMU block and updates the SPMU every clock cycle. This continues till the SPMU is populated with all the $4 \times 4,096$

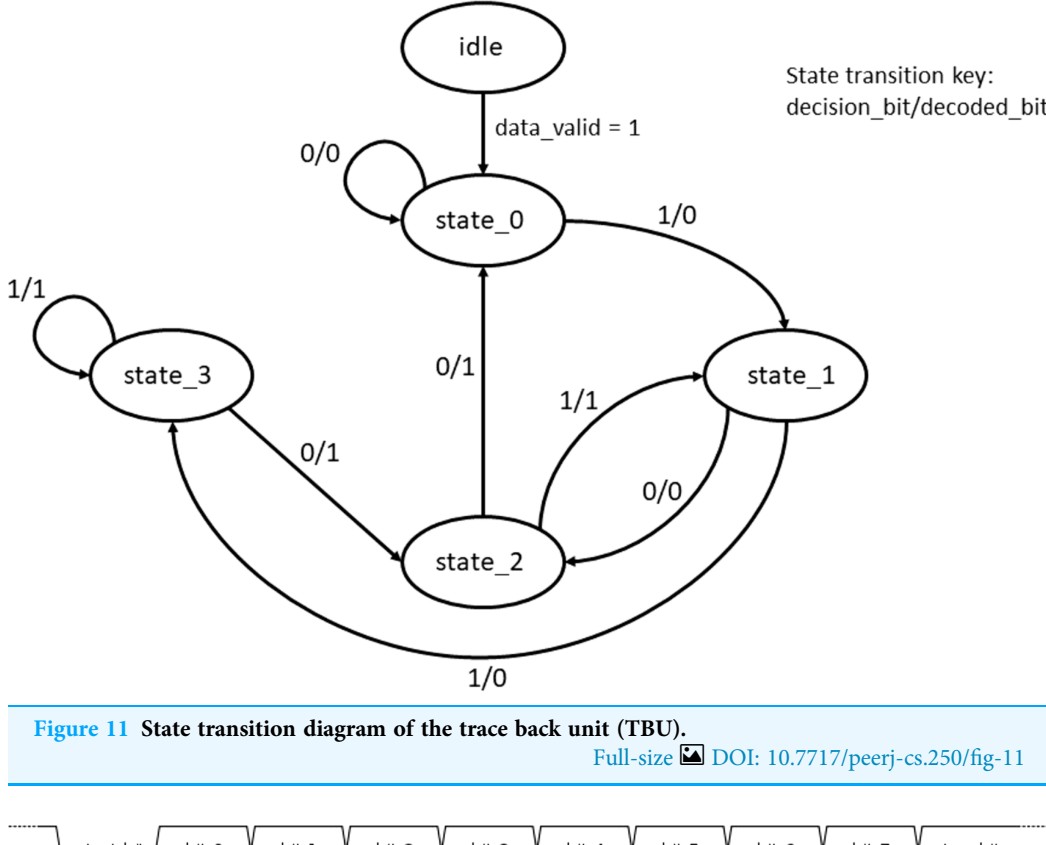

**Figure 11 State transition diagram of the trace back unit (TBU).**

| start bit | bit 0 | bit 1 | bit 2 | bit 3 | bit 4 | bit 5 | bit 6 | bit 7 | stop bit |

**Figure 12 10 bit UART frame.**     

decision bit values. Once the SPMU is populated with the decision bits, the decode_en bit is triggered high and the trace back unit kicks in to start the reverse process to decode the original input bits. Figure 11 shows the state machine used for the trace back operation.

Since the encoded message is terminated with zeros, the final state of the system would be the zero state and hence TBU can be safely initialized to start the decoding from the zero state. After decoding all the 4,096 bits, decode_done signal is activated to trigger the output interface. It is to be noted that it takes around 4,096 clock cycles for the forward tracing, 4,096 clock cycles for traceback and few cycles are required for synchronization.

### Output interface

The output interface consists of a first-in-first-out (FIFO) buffer of size $8 \times 512$ (4,096 bits) to store the decoded data and the baud generator and transmitter modules of UART to transmit the data serially on to a PC. The UART is designed with a frame length of 10 (one start bit, eight data bits and one stop bit) and works at a baud-rate of 9,600 bps. Figure 12 shows the UART frame used in the design.

The data from the decoder is first stored in the FIFO and is transmitted when an external request for data transfer is enabled. The FIFO is designed using a BRAM of size $8 \times 512$. FIFO is chosen to be one byte wide to enable byte wise data transfer to the UART

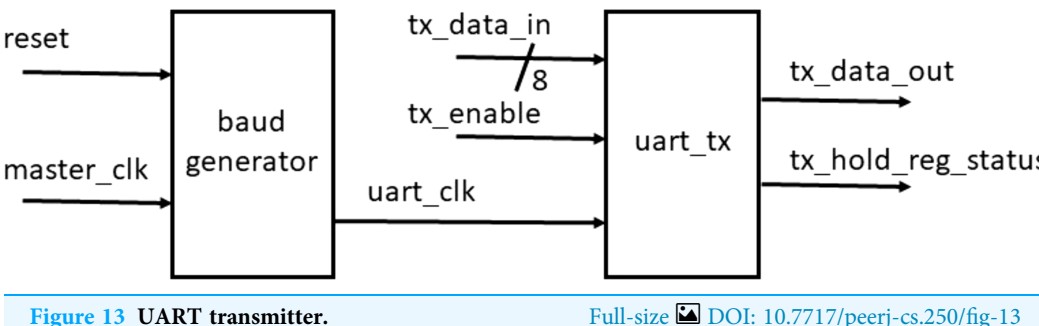

**Figure 13  UART transmitter.**

transmitter easily. Figure 13 shows the transmitter block designed to operate at a baud rate of 9,600 bps.

The UART transmitter contains the baud generator that generates the UART clock corresponding to a baud rate of 9,600 bps. To generate this clock from the master clock, we use a clock divider whose value can be obtained by using Eq. (1)

$$\text{Baudrate} = \frac{\text{master\_clk}}{(16 \times \text{divisor})} \tag{1}$$

Since the master clock of the FPGA board (Zybo-Z7010 ) is 125 MHz, we need to design a clock divider of value given by Eq. (2)

$$\text{divisor} = \frac{\text{master\_clk}}{(16 \times \text{Baudrate})} = 813.4 \tag{2}$$

Rounding off to the nearest highest integer, we use a clock divider of value 814 to generate the UART clock and data is transmitted out serially to the PC through a RS232 interface.

# SIMULATION AND IMPLEMENTATION OF VITERBI DECODER

## Input data generation

A coded communication system is set up as detailed in the introduction section, using MATLAB. 4 K message bits are generated randomly and encoded using a convolutional encoder of rate 1/2 to generate 8 K encoded bits. Binary phase shift keying modulation is used to modulate the encoded message stream and the resulting data sequence is transmitted through an additive white gaussian noise channel of varying signal to noise rations (SNRs). The received sequences are demodulated and stored as inputs for the Viterbi decoder.

## Design and functional verification

The Viterbi decoder was first designed using a behavioral model in MATLAB and then a synthesizable RTL design was done in Verilog. The original encoded message sequence (without noise addition) is given as input to the decoders and the output was verified to be the original message sequence. This establishes that the decoders are functionally correct.

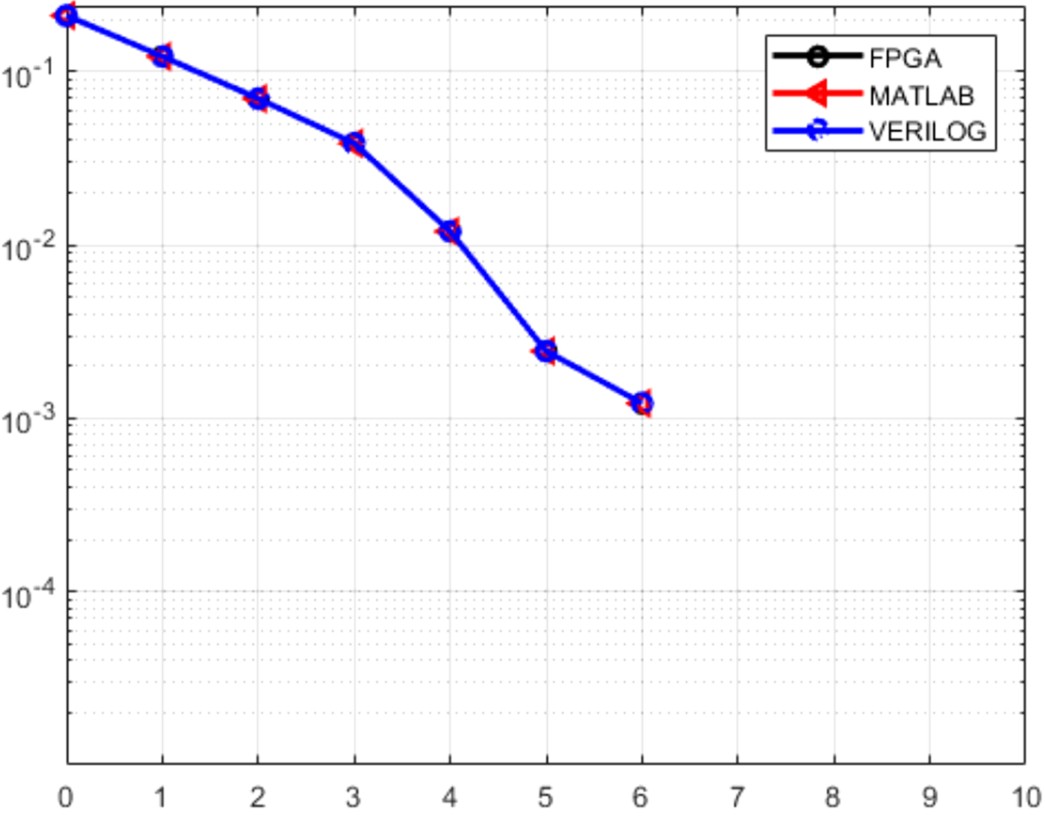

**Figure 14 Comparative BER plots for MATLAB behavioral design, RTL design and the FPGA implementation (beyond 6 dB, the BER is zero).** All three plots overlap perfectly, thus establishing the correctness of the implementation.               

Having verified the functionality of the designs using the encoded message as the inputs, the noise-added sequences are given as inputs and the BERs are computed.

## FPGA implementation

The RTL design of the decoder was synthesized on to a Zybo board that is built using Z-7010, a member of Xilinx Zynq-7000 family. Z-7010 is designed using Xilinx all programable system-on-chip architecture, that integrates a Xilinx 7-series FPGA along with an ARM based Cortex-A9 processor. In our work, we have not used the ARM processor for generating test inputs but instead, the test vectors are generated from MATLAB as briefed above and a single port BRAM is initialized with these input test vectors. The decoded output bits are first stored in another BRAM and then sent by the UART transmitter to the PMOD pins of the Zybo board. This transmitted data is driven through a UART-to-USB translator (PL2303) and the serial bits are captured on the PC using Real Term serial terminal (*RealTerm, 2019*).

Functional verification of the implemented design was done in the same manner as the simulated design. The encoded message bits without noise addition were given as the input and the decoded bits were compared with the original message sequence. It was

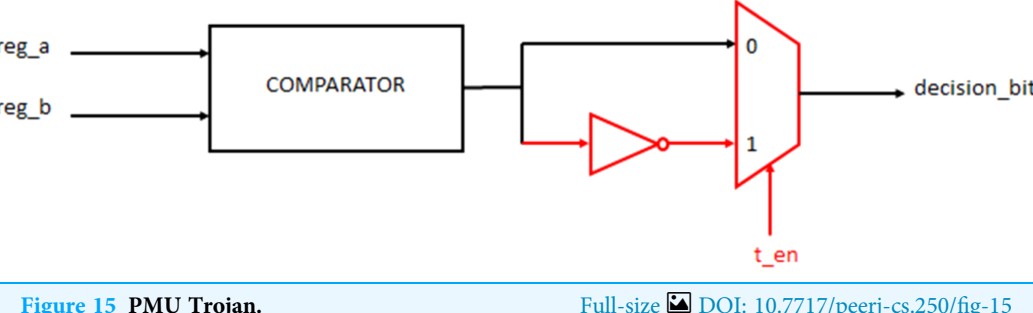

**Figure 15  PMU Trojan.**       

verified that the decoded and input message bits matched successfully. For calculating the decoder performance, the decoder was fed with noisy data of different SNRs.

Figure 14 shows the BERs obtained for the MATLAB behavioral model, the RTL design and the FPGA implementation where it can be seen that the BER drops down to zero for SNRs greater than 6 dB[1]. Also, there is a perfect overlap of all the plots, thus demonstrating the equivalence of the behavioral, RTL and the implemented models.

## TROJAN DESIGN AND IMPLEMENTATION

In this work, we propose the design of three possible Trojans and study how their stealthy presence may affect the system performance.

### Trojan design 1: decision-bit flipping Trojan (PMU Trojan)

In the PMU, it is expected that the comparator identifies the least path metric path and correspondingly store a "1" or "0" to indicate either of the paths to be traversed during trace back. The proposed Trojan Decision-bit flipping, when enabled, flips the decision bits thus causing the trace back unit to proceed in an incorrect decoding path. The hardware model of the Trojan circuit is shown in Fig. 15.

The decision bit is inverted when Trojan is enabled and the SPMU gets populated with an incorrect value. This causes the lower metric path to be discarded instead of the higher path metric, thus resulting in possible erroneous decoding.

### Trojan design 2: traceback path modification Trojan

During trace back, the decision bits from the SPMU is read and a path is chosen based on whether the stored value is a "0" or a "1". The traceback path modification Trojan inverts this logic and changes the state transitions, thus making it proceed in the wrong path. Figure 16 shows the modified state machine due to the Trojan being effective.

When the Trojan is enabled, the transitions are made to differ from the original state transitions resulting in erroneous decoding.

### Trojan design 3: shift-direction-modifying Trojan

In the output interface, when the decoded data is being written into the shift register, normally it will be right shift operation. But when the Trojan is enabled this operation will be reversed and performs left_shift thus sending erroneous data to the transmitter. This can be achieved by the use of multiplexers that can alter the shift direction based on

[1] Since the BER is plotted on a log scale, it is not possible to indicate the zero values on the plot

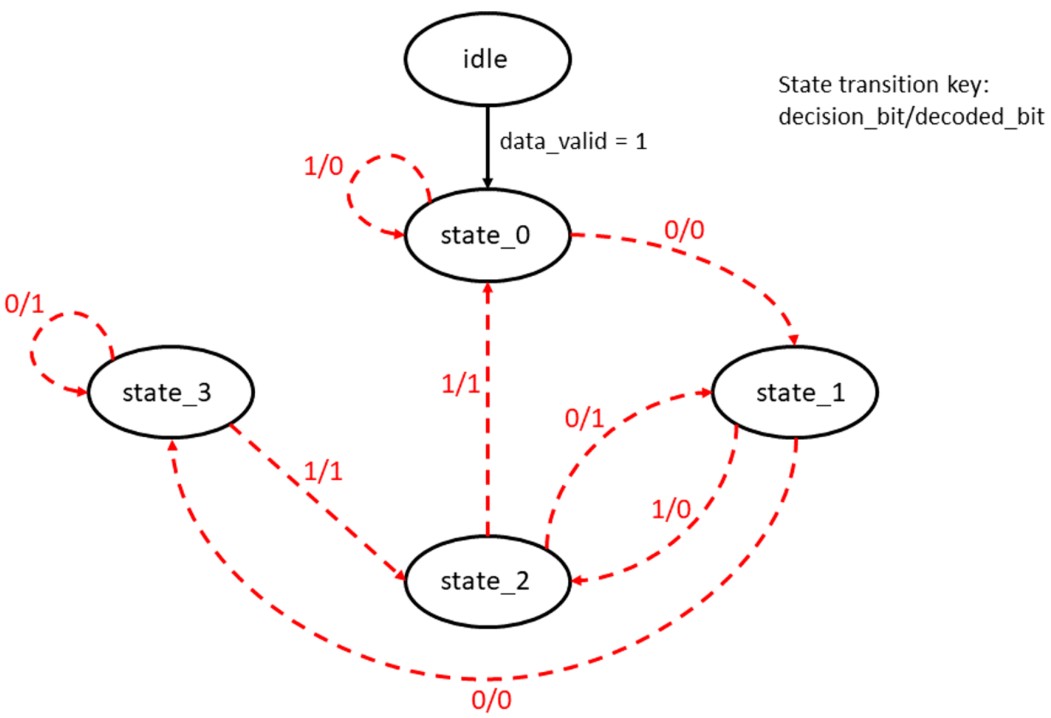

**Figure 16 Trojan effect modifying the trace back path in the TBU.**

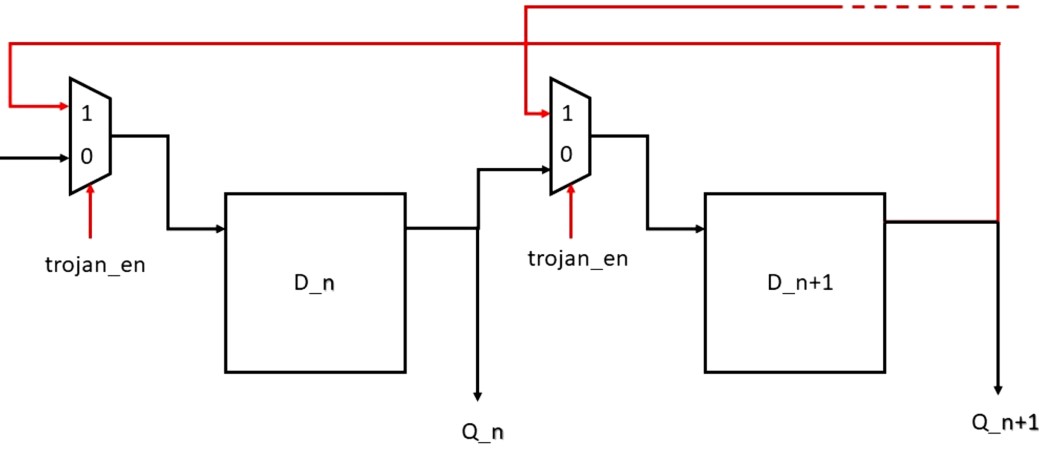

**Figure 17 Shift direction modifying Trojan.**

the Trojan enable signal. Figure 17 shows the Trojan circuitry that is created due to this Trojan.

## RESULTS AND DISCUSSIONS

The effectiveness of the designed Trojans can be gauged by their stealthy nature and by their propensity to degrade the performance of the infected system.

**Table 1 Power summary table.**

| Power summary (W) | Without trojan | PMU trojan | TBU trojan | Shift modification trojan |
|---|---|---|---|---|
| Total on-chip power | 0.129 | 0.129 | 0.133 | 0.13 |
| Dynamic | 0.025 | 0.025 | 0.03 | 0.026 |
| Device static | 0.104 | 0.104 | 0.104 | 0.104 |

**Table 2 Utilization summary table.**

| Utilization | | Without trojan | | PMU trojan | | TBU trojan | | Shift modification trojan | |
|---|---|---|---|---|---|---|---|---|---|
| Site type | Available | Used | Util% | Used | Util% | Used | Util% | Used | Util% |
| Slice LUTs | 17,600 | 453 | 2.57 | 451 | 2.56 | 457 | 2.59 | 455 | 2.58 |
| LUT as logic | 17,600 | 197 | 1.11 | 195 | 1.1 | 201 | 1.14 | 199 | 1.13 |
| LUT as memory | 6,000 | 256 | 4.26 | 256 | 4.26 | 256 | 4.26 | 256 | 4.26 |

## Stealthy Trojans

Trojans by nature are stealthy in nature and are difficult to detect. To verify if the proposed Trojan models are stealthy enough to evade detection, the difference in the area and power dissipated due to the Trojan insertion are calculated. Tables 1 and 2 show the power and utilization summary of the Trojan free circuits and the Trojan affected circuits. It is to be noted that the parameters obtained is only for the decoder logic without including the input BRAM and the output UART.

It can be seen from the power and area utilization results that in the worst case, there is a difference of only 4 mw of on-chip power (difference of 3.1%), and only four extra LUTs (change of 0.5%) due to the Trojan insertions. This establishes its stealthy nature, thus qualifying them as effective Trojans.

## Performance degradation

To analyze how effectively the Trojans disrupt the natural decoding process, the Trojans are triggered at random time instants and the decoded bits are analyzed. Generally, Trojans are designed to activate surreptitiously in order to go unnoticed. Hence the triggering was done only once during the entire decoding process and the effect of this triggering is observed and the resultant BER is calculated.

### Trojan triggering logic

Figure 18 shows the circuit for generating Trojan enable signal. It consists of a BRAM, a 14 bit counter to count up to the maximum possible number of clock cycles required for decoding and a comparator. The Trojan can be triggered any time during the entire duration of decoding. To identify these triggering instances, 50 random numbers are generated and stored in a block RAM. During each triggering one location is read from the BRAM as the triggering instance. The Trojan enable signal is generated when the counter value matches with the random number being read from the BRAM. The BER is calculated to be the average of the BERs obtained from all the triggering instances.
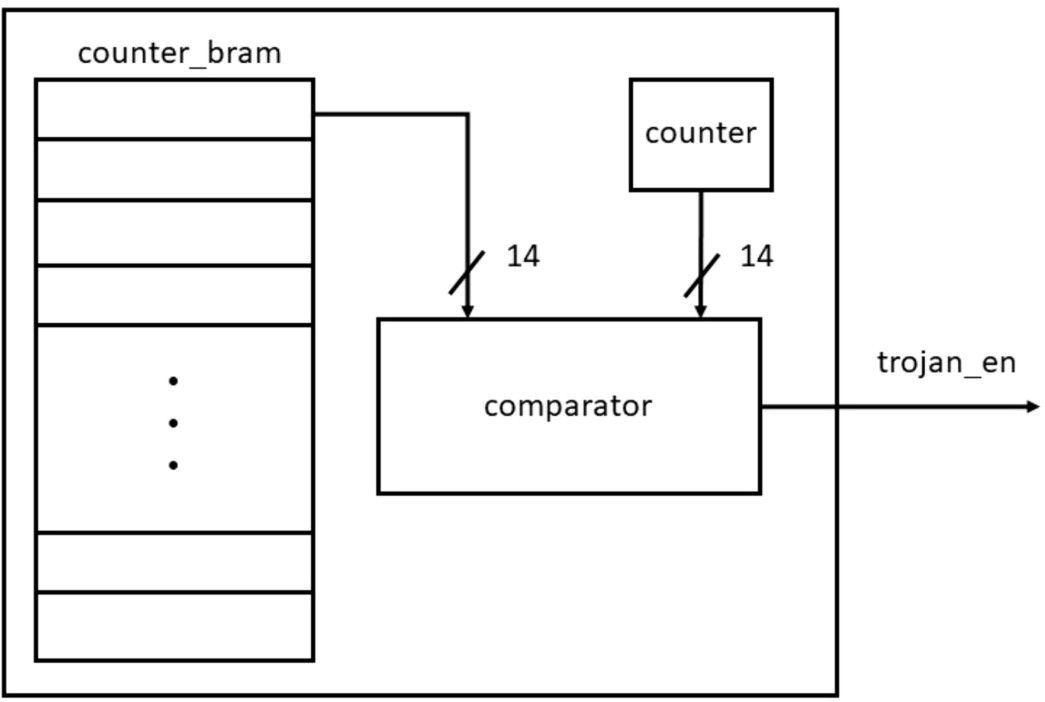

**Figure 18 Trojan triggering logic.**

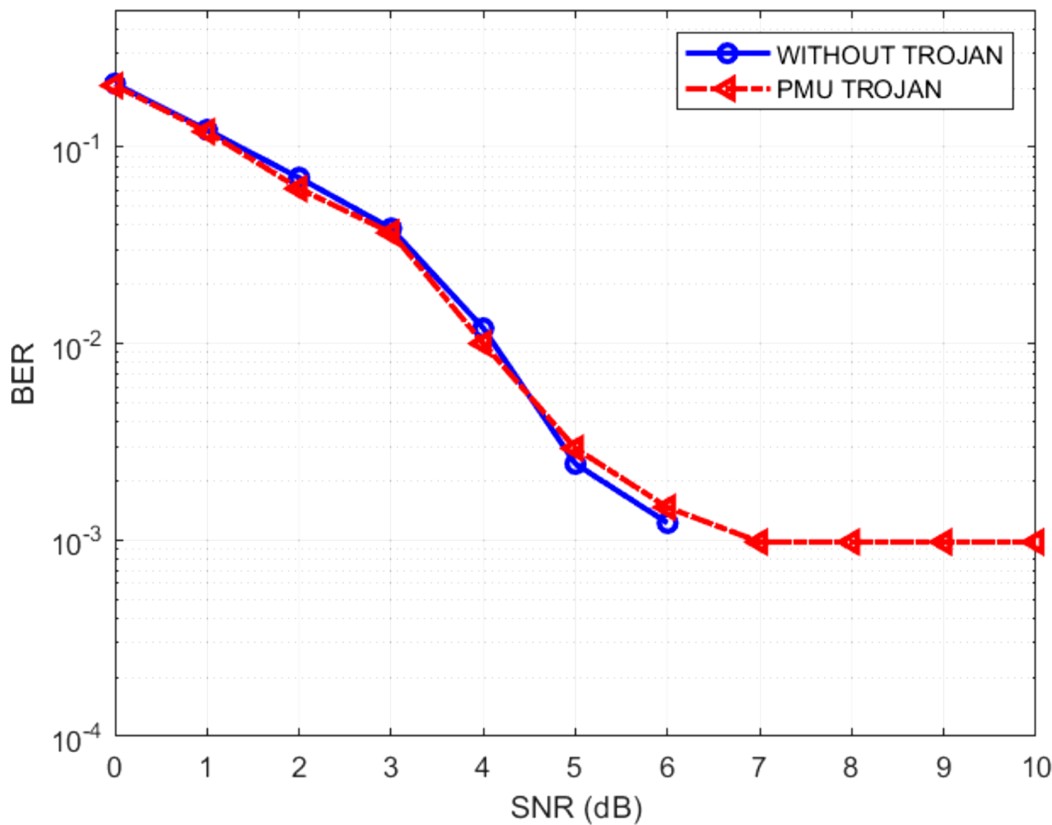

**Figure 19 BER plot for PMU Trojan.**

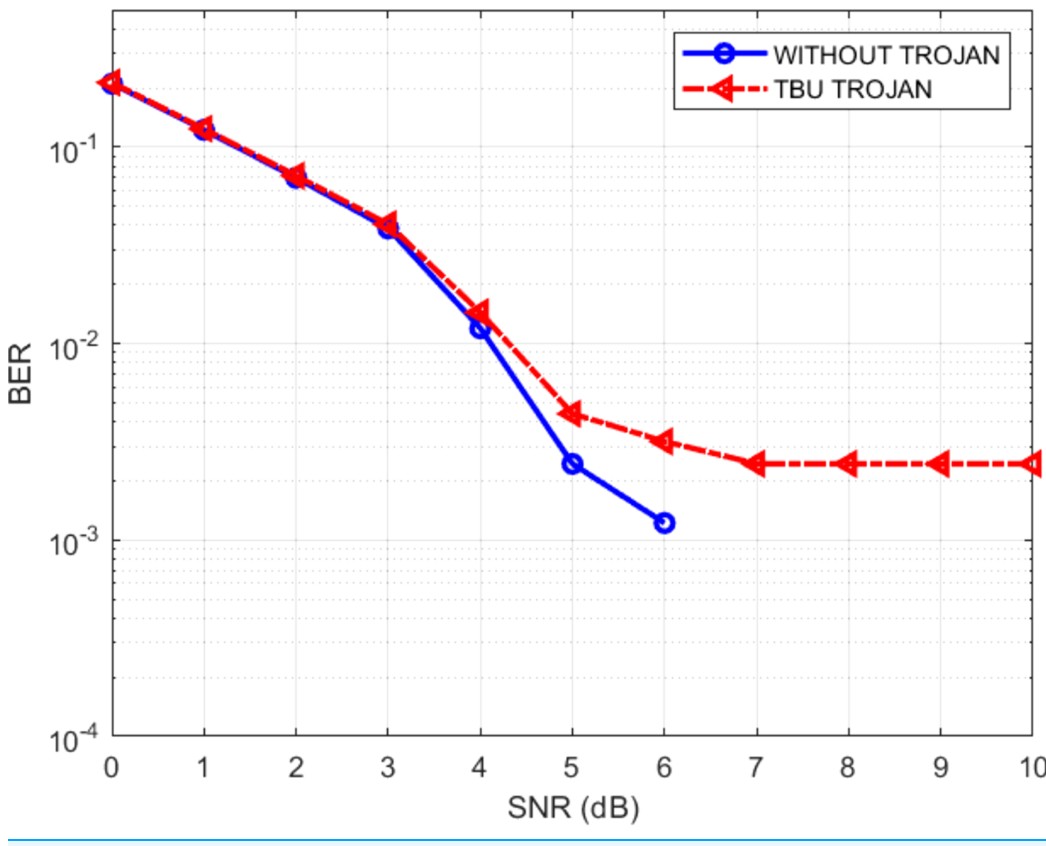

**Figure 20  BER plot for TBU Trojan.**     

### Effect of the triggered Trojans

The effect of the Trojans may be quantified by the increase in the BERs of the infected decoder. Figures 19–21 show the comparative BER graphs of the decoder without Trojan and with Trojan being activated only once.

It can be seen that in the absence of Trojans, the BER drops down to zero for SNRs greater than 6 dB, but with the Trojans being active, the BER doesn't reduce to zero even for high SNRs. Thus the Trojans leave a distinctive BER signature (high BER). Among the three Trojans, the BER signature is highest for TBU Trojan and lowest for PMU Trojan, with the shift modification Trojan producing a BER signature in between the other two.

It is also interesting to note that the difference in the performances between the Trojan free design and the design with Trojans is negligible in the low SNR regions but the difference is prominent in the high SNR regions. Also at some low SNR conditions, the performance of the Trojan affected system is slightly better than the unaffected system. This scenario is possible since, in the low SNR regions, the data itself is noisy and erroneous. During the bit flipping or the state transition or the shift direction modifying actions of the Trojans there exists a possibility that few erroneous bits are converted to correct bits, thus providing a reverse effect on the system. The possibility of this kind of behavior, along with the fact that Trojans are stealthy make it difficult to conjure

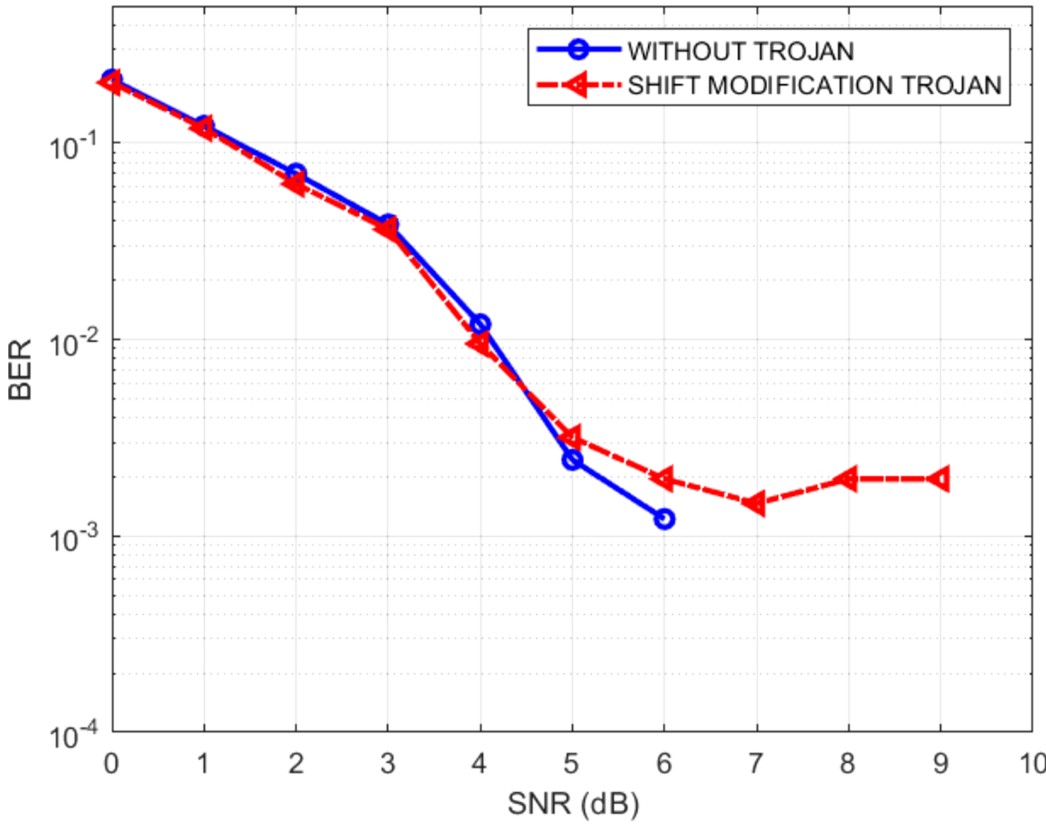

**Figure 21 BER plot for shift direction modifying Trojan.**

effective Trojan detection schemes. To counter such situations, the current focus of researchers is to neutralize them apart from detecting the Trojans (*Gunti & Lingasubramanian, 2017*).

## Study limitations

The study proposes stealthy Trojans—*decision bit flipping, traceback path modification and shift direction changing* Trojans—and their effect on the decoding efficiency of a Viterbi decoder. The Trojans degrade the performance of the decoder, causing it to have a high BER. But, it is to be noted that high BER can also arise in a system due to high noise in the channel. Hence in situations where the channel's noise characteristics are unknown, the presence of these Trojan can't be inferred purely from the BER signature. It needs to be augmented with other Trojan detection schemes to correctly infer the presence of Trojans.

## CONCLUSIONS

In this work, we have designed a FPGA based implementation of a Viterbi decoder and presented possible effects of hardware Trojans on coded communication systems. Three unique threat models are developed and tested on the Viterbi decoder which is popular

for its low BER performance. However, we show that the presence of the proposed Trojans affect the efficiency of the Viterbi decoder by increasing the BER. The stealthiness of the proposed Trojans is also established. Using the proposed threat models, we envision to test their effects on complex systems like CPS and IoT which rely on efficient communication channels.

With the wide application of convolution codes in various SNR scenarios, the results of the implemented system play a significant role in emphasizing the need for efficient Trojan detection schemes. It is envisaged that apart from BER signature analysis, other Trojan detection and neutralizing schemes will be explored for the proposed Trojans.

### Funding
This work was supported by Space Application Center, ISRO through RESPOND project /ISRO/RES/3/732/16-17. Deepak Mishra from ISRO is a coauthor and was involved in the study design, analysis and preparation of the article.

### Grant Disclosures
The following grant information was disclosed by the authors:
Space Application Center, ISRO through RESPOND Project: /ISRO/RES/3/732/16-17.

### Competing Interests
Deepak Mishra is a scientist at the Indian Space Research Organization (ISRO), Ahmedabad, India.

### Author Contributions
- Varsha Kakkara conceived and designed the experiments, performed the experiments, analyzed the data, performed the computation work, prepared figures and/or tables, authored or reviewed drafts of the paper, and approved the final draft.
- Karthi Balasubramanian conceived and designed the experiments, performed the experiments, analyzed the data, prepared figures and/or tables, authored or reviewed drafts of the paper, and approved the final draft.
- B. Yamuna conceived and designed the experiments, analyzed the data, prepared figures and/or tables, authored or reviewed drafts of the paper, and approved the final draft.
- Deepak Mishra conceived and designed the experiments, authored or reviewed drafts of the paper, and approved the final draft.
- Karthikeyan Lingasubramanian conceived and designed the experiments, analyzed the data, authored or reviewed drafts of the paper, and approved the final draft.
- Senthil Murugan conceived and designed the experiments, performed the experiments, analyzed the data, performed the computation work, authored or reviewed drafts of the paper, and approved the final draft.

## Data Availability

The raw data generated from MATLAB are available in the Supplemental Files.

## Supplemental Information

Supplemental information for this article can be found online at http://dx.doi.org/10.7717/peerj-cs.250#supplemental-information.

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
