# Peer review of "A Viterbi decoder and its hardware Trojan models: an FPGA-based implementation study"

_PeerJ Computer Science, doi:10.7717/peerj-cs.250_

## Round 0.1 · original submission · Major Revisions

All of the reviewers have recommended major revisions to your paper.

Reviewer 1 ·

Basic reporting

The article is well written and contains the appropriate amount of information. However a thorough literature review / background section would be welcomed. This would allow the reader to position the paper within the field and understand what was done before and what are the contributions of the paper.

As a broad example, the work discusses a 3% increase in heat due to the trojan, a brief search on google scholar discusses the detection of trojan using off-the-shelf work to detect trojan using heat cameras.

Experimental design

The paper provides information about the design and experiment, and while the authors provide information about the creation of the trojan, other papers have been discussing the creation of trojan for this type of work and it would be good to see the decision process taken throughout the paper.

Furthermore, the authors should clarify their decision of designing the trojans themselves rather than using known trojan as the performance degradation are interlinked with the design and choices made for the trojans.

Validity of the findings

Generally, the paper provides a good overview and description of the work, however the authors should provide a longer discussion on the effects of the performance degradation due to the type of trojan being used to disrupt the initial design.

The conclusion is well written and provides enough information to summarise the work undertaken However, the authors should identify some future work as well as the limitations of their work.

·

Basic reporting

This paper presents an FPGA implementation of Viterbi Decoder and tests the effects of hardware Trojans on coded communication systems. There needs to be a solid connection between the introduction of the hardware trojans and the actual trojans that this paper implements.

Experimental design

One question of the experimental design is that in figure 20 to 22, why in lower SNR, the BER is even lower with trojans?

Second is that the zip file provides not much directly useful message. Find another way of delivering the experiment result.

Validity of the findings

One suggestion is that more newer and closely-related works are needed to be explored and the direct comparison is needed.

Additional comments

Has the potential to be accepted if solved the above-mentioned issues.

·

Basic reporting

Manuscript is quite comprehensible. References are appropriate.

Background material on hardware trojans is
adequate. There are minor grammatical
issues -

Example: line 35: "A hardware of a communication system" to "The hardware of a.."
line 36: "This will allow.. " to "This allows"
Please add a paragraph to explain Figure 1 that describes
classification of hardware trojans.

Authors also assume familiarity with viterbi decoding.
For example: line 99, it is unclear what
"hard decision metric" and "cumulative branch metrics" refer to.
"The decoder computes the hard decision metric for each branch starting
from the zero initial state. The cumulative branch metrics along with all the possible paths leading to a node or state constitutes the path metric.
Consider elaborating the "viterbi decoder" section with a simple example to improve clarity.
line 100: At any node the path with the lowest path metric is retained as the survivor path and the
101 rest of the paths are discarded
Please add appropriate punctuation.

Tables and Figures appear adequate.

Experimental design

This work describes a technique to detect the presence of hardware trojans
in communication systems by analyzing bit error rate (BER) of the decoded
data stream. Detecting hardwrae trojans is a difficult problem when
the malicious circuit is designed with small area/power footprint
and therefore, the presented research problem is valid.

The paper assumes trojan is always inserted at the decoder. Why?
Can trojns inserted at the encoder be detected? and if yes how?
Please explain in the introduction why the paper focuses on trojans inserted
at decoder.

Please make it clear how this work is different from your prev. conference submission.

Authors have provided a thorough illustration of experimental setup, decoder block diagram etc. enough to reproduce results.

The main conclusion from this work appears to be that hardware trojans
have a distinct BER signature (high BER). But, the presence
of this signature doesn't guarantee the existence of a hardware
trojan. For example, high noise in the channel can also
result in high BER which raises the question - if
a channel's noise characteristics are unknown,
how do you conclude the presence
of a trojan purely from BER analysis?

Validity of the findings

Results need more data and explanation.

1. line:171 mentions the BER drops to 0 for SNR > 6db. But, the plot in Fig 14
doesn't report BER for SNR > 6dB. Also, BER for 6dB
is shown as 10^-4 which isn't zero. Please provide complete
data and fix description to match the plot.

2. Plots 14 and 15 can be combined to better illustrate that FPGA measurements
match matlab model.

3. Figure 20 is incomplete since it doesn't report BER results without trojan for SNR > 6dB.
Please fix.

Additional comments

Overall, well written paper.

---

## Round 0.2 · Minor Revisions

There are just a few minor things to clean up before this paper can be accepted. Please address the issues the reviewers raise. Thank you.

·

Basic reporting

Grammatical issues have been addressed. References are appropriate.
Please remove references to supplemental figures, unless they may be included in the final version of the manuscript.

Experimental design

Concerns have been addressed.

Validity of the findings

No comment

Additional comments

My concerns have been addressed. Please remove references to supplemental figures, unless they may be included in the final version of the manuscript.

Reviewer 4 ·

Basic reporting

This paper demonstrates the implementation of hardware trojan in a Viterbi decoder implemented by FGPA. They have shown the changes in LUT used and power dissipation with trojan inserted. Meanwhile, they also demonstrated the BER drops with the presence of Trojans.

Strengths:
1. The author gives an overview of the hardware trojans and more specifically how the trojans work in the channel decoders.
2. This paper tests the effects of BER dropping with the triggered Trojans.

Experimental design

However, I think the contributions and the novelty of this paper are not very promising and clear to me.

1. First of all, from the title and abstract, I get the main theme for this paper is discussing how hardware Trojan attacks are vulnerable to the Viterbi decoder. However, the structure of this paper makes the theme not clear. This paper uses more than half of the contents to illustrate the process in designing the Viterbi decoder, which I don’t think it aligns with the main theme of the title. I admit that to add trojans, one needs to be 100% clear on the RTL structure of the design, however, too much details on the design but with less than 1/3 on the discussion of the hardware trojan makes the structure of the paper problematic. To make this paper logic acceptable, I would suggest the author shrink the part discussing the design of Viterbi decoder but with emphasis on the hardware trojan.

2. The author demonstrates the effectiveness of the hardware trojan on the designed Viterbi decoder. I wonder whether the trojan the author proposed are fixed to those structures or they can be universally used in the other structure of Viterbi decoder? If the proposed trojans are just applied on the proposed structure, the work of this paper should be improved by proposing some universal structure which works at least for all the Viterbi decoder.

3. This paper has some evaluation results on the hardware trojan and performance analysis on the BER drops. However, I think the author might need to add more discussion on the security analysis of the hardware trojan. For example, the author would like to discuss how the triggering success rate for each trojan. Also, it lacks to distinguish the influence of the three different kind of trojan on the performance of the BER drops. Moreover, it would be sufficient if the author evaluates the trojan detection method on his proposed trojans to further confirm that the trojans are feasible and not easy to be detected.

Validity of the findings

Some editorial remarks.
The author should proof-read the whole paper and modify some mistakes in the format. A few examples are incorporated but not an all-inclusive list.
1 Please scale figure 11 and figure 14.
2 please use the same font size for the Table 1 and Table 2. The size of the column “shift modification trojan” is larger than the other columns.

---

## Round 0.3 · accepted · Accept

This version of the paper has addressed the reviewers concerns.